# Scaling Analysis of Interleaved Speech-Text Language Models

**Gallil Maimon,    Michael Hassid,    Amit Roth,    Yossi Adi**
Department of Computer Science and Engineering
Hebrew University of Jerusalem
gallilmaimon@mail.huji.ac.il

## Abstract

Existing Speech Language Model (SLM) scaling analysis paints a bleak picture. It predicts that SLMs require much more compute and data compared to text, leading some to question the feasibility of training high-quality SLMs. However, modern SLMs are often initialised from pre-trained TextLMs using speech-text interleaving to allow knowledge transfer. This raises the question - *Do interleaved SLMs scale more efficiently than textless-SLMs?* In this paper we answer a resounding *yes!* We conduct scaling analysis of interleaved SLMs by training several dozen and analysing the scaling trends. We see that under this setup SLMs scale more efficiently with compute. Additionally, our results indicate that the scaling dynamics significantly differ from textless-SLMs, suggesting one should allocate notably more of the compute budget to increasing model size over training tokens. We also study the role of synthetic data and TextLM model families in unlocking this potential. Results suggest that our scaled up model achieves comparable semantic speech performance to leading models, while using less compute and data. We open source models, samples, and data - https://pages.cs.huji.ac.il/adiyoss-lab/sims/.

## 1 Introduction

Speech Language Models (SLMs) have gained much attention from the research community (Cui et al., 2024a; Huang et al., 2024; Maimon et al., 2025b) showing impressive results in reasoning over speech and audio (Chu et al., 2023; Tang et al., 2023), and as a basis for spoken chat systems (Défossez et al., 2024; Ji et al., 2024). Early efforts in constructing SLMs were focused on training with speech data only (Lakhotia et al., 2021; Borsos et al., 2023), often known as textless-SLMs. Such approaches showed potential in speech modelling and generation, yet, such modelling approach was found to be limited in semantic abilities.

Recent methods for training SLMs integrate text alongside speech, either by processing both modalities concurrently (Défossez et al., 2024; Fang et al., 2024) or by interleaving between them (Nguyen et al., 2024; Zeng et al., 2024). These models are often referred to as joint speech-text SLMs. Such modelling approach resulted in significant improvement in SLM performance, specifically considering semantic content understanding and generation. Despite the great results, the scaling properties of such training paradigms remain unclear.

Following the scaling analysis of TextLMs (Hoffmann et al., 2022; Kaplan et al., 2020), Cuervo & Marxer (2024) proposed the first scaling analysis for textless-SLMs, focusing on the Generative Spoken Language Models (GSLM) approach (Lakhotia et al., 2021). The authors trained models of different sizes and token counts to fit a parametric function which describes the validation loss as a function of the model parameters, and training tokens. Their analysis suggests that the semantic abilities of SLMs scale notably slower than TextLMs and would require significantly larger datasets for training ($\sim 3X$ more than text). While thorough and valuable to the speech community, the analysis presented by Cuervo & Marxer (2024) centres on textless-SLMs and overlooks the role of the text modality and its impact on the scaling behaviour of joint speech-text SLMs.

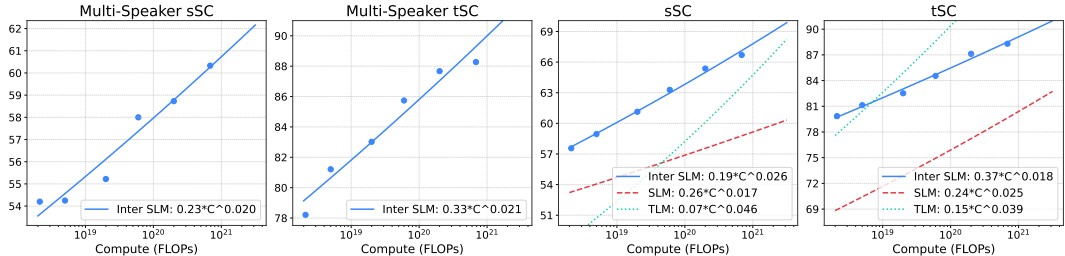

Figure 1: Analysing scaling of interleaved SLMs, considering the best model size per-compute, across semantic speech metrics - Topic StoryCloze and Semantic StoryCloze (Hassid et al., 2023). We compare scaling trends to textless SLMs (denoted SLM), and TextLMs (TLM), as presented by Cuervo & Marxer (2024).

In this work, we explore the scaling behaviour of joint speech-text SLMs within an interleaved setup. We train multiple interleaved SLMs across a range of sizes, compute budgets, and model families to derive practical insights for optimising SLM performance. Specifically, after identifying well-performing model families and data mix - we fix several compute budgets and compare models of different sizes trained with the same compute budget.

Compared to textless-SLMs, evaluating the scaling properties of interleaved SLMs poses further challenges. For instance, initialisation from a TextLM could have a key impact on results (Hassid et al., 2023), but not explicitly modelled in scaling. Furthermore, since there are different modalities in training it is unclear with regards to which tokens one should analyse the scaling. While we do not solve all in this paper, we shed light on these challenges, and propose practical analysis methods. These provide insights into how to best allocate compute between model parameters and tokens, as well as the potential benefits of scaling, as shown in Figure 1.

The insights found directly translate into better SLMs, which far out-perform the expected results when considering speech-only scaling laws (Cuervo & Marxer, 2024). This in turn gives an optimistic tone to the feasibility of training high-quality SLMs with existing compute and datasets. We apply all insights and train a 7B interleaved speech-text LM which achieves performance comparable to leading SLMs considering semantic speech metrics. We open source the above models and training code, thus encouraging the community to further analyse such scaling properties, and text-speech knowledge transfer.

## 2 Background

### 2.1 Speech Language Models

The speech research community has explored various types of SLMs (Cui et al., 2024b). In this work, we focus on models trained using both speech and text tokens, commonly referred to as joint speech-text SLMs. This approach typically involves three key components: (i) converting speech to discrete units, (ii) a joint speech-text SLM, and (iii) converting units back to speech, with each module trained independently. While previous studies introduced different methods for jointly modelling speech and text, we focus on modality interleaving.

**Speech-to-unit** modules encode the raw speech signal into a discrete sequence. The common approach first encodes speech into a continuous representation and then quantises it to get a sequence of discrete units. Denote the domain of audio samples by $\mathcal{X} \subset \mathbb{R}$. The representation for a signal is therefore $x = (x_1, \ldots, x_T)$, where $x_t \in \mathcal{X}$ for all $1 \le t \le T$.

Let us consider an encoder network, $f$, that takes a speech utterance as input and produces a sequence of lower-frequency representations: $f(x) = (v_1, \ldots, v_{T'})$. While previous studies have examined various speech encoders, $f$, our focus is on HuBERT (Hsu et al., 2021) as it is the most widely used speech encoder for SLMs. Because the outputs of this model are continuous, a k-means clustering is used to convert the $f(x)$ into discrete units, denoted as

$z = (z_1, \ldots, z_{T'})$. Each $z_i$ represents a discrete category, with values in the range $z_i \in 1, \ldots, K$ for $1 \le i \le T'$, where $K$ is the total number of discrete units.

**Interleaved speech-text SLMs** are trained on a combination of discrete speech units $z$ and text tokens $t$ by mixing them together in sequence. Given time-aligned transcriptions $(W_i, Start_i, End_i)$ that specify when each word begins and ends time-stamps, we assign a modality token $M$ (either speech or text) to each segment. By grouping consecutive words with the same modality, we create modality-specific time spans within an utterance, such as: $[(Text, 0, 1.3), (Speech, 1.3, 3.5), \ldots]$. Each segment is then tokenized separately, producing an interleaved sequence like $s = (Text, t_1, \ldots, t_m, Speech, z_k, \ldots, z_l, Text, t_j, \ldots)$. For example, an interleaved phrase might look like: "[TEXT] the cat [SPEECH] [Hu3] [Hu7] ... [Hu200] [TEXT] the mat".

Different approaches to assigning modalities to time-spans or words have been suggested. We adopt a strategy similar to that of Zeng et al. (2024), where speech segments are chosen by sampling their lengths (in words) from a Poisson distribution with $\lambda = 10$, continuing until these segments make up $\eta = 0.3$ of the total word count. A language model (often initialised from TextLM; Hassid et al. 2023) is then trained on the resulting mixed-modality sequences following the standard next-token prediction objective. Previous works have also trained models on individual modalities - i.e. separate speech-only and text-only datasets.

**Unit-to-speech** module converts the speech discrete units to a raw waveform. We follow, Polyak et al. (2021); Lakhotia et al. (2021) in using a unit-based vocoder based on the HiFi-GAN architecture to convert units to speech directly.

## 2.2 Scaling Analysis

Scaling analysis in LMs examines how performance scales with increased compute resources (mainly training FLOPS, denoted $C$). This is mainly characterised by model size ($N$) and training data size ($D$), and is commonly approximated as $C \approx 6ND$ (Kaplan et al., 2020). Several works indicate that final model loss ($L$) follows an approximate power-law relationship with respect to $N$, $D$ and consequently $C$ (Clark et al., 2022; Kaplan et al., 2020):

$$L(N) \propto N^\alpha; \quad L(D) \propto D^\beta; \quad L(C) \propto C^\gamma.$$

Building on this, Hoffmann et al. (2022) proposed three approaches for scaling analysis of language models. The first fixes model sizes and varies the number of training tokens. The second aims to fit a parametric function to estimate the loss as a function of $N$ and $D$:

$$\hat{L}(N, D) = E + \frac{A}{N^\alpha} + \frac{B}{D^\beta},$$

where $E$, $A$, $B$, $\alpha$, and $\beta$ are parameters empirically estimated using data from multiple trained models. The third approach varies the model size for a fixed set of different training FLOPs. It then identifies the optimal model size for each training FLOPs budget. Based on the optimal models, it fits a function to predict the optimal number of training tokens or model size given training FLOPs budget. This approach was termed ISO-FLOP curves.

## 3 Experimental Setup

We start by describing our experimental setup considering: (i) dataset used for model training; (ii) optimisation and implementation details; and (iii) model evaluation.

**Data.** We follow Cuervo & Marxer (2024) using a collection of diverse, English datasets, namely - LibriSpeech (Panayotov et al., 2015), LibriLight (Kahn et al., 2020), VoxPopuli (Wang et al., 2021), Tedlium (Hernandez et al., 2018), People Speech (Galvez et al., 2021) and SWC (Baumann et al., 2018) for training. We also utilise a single speaker synthetic dataset sTinyStories, which was shown to improve semantic abilities in non-interleaved SLMs (Cuervo & Marxer, 2024; Maimon et al., 2025a). Time aligned transcriptions for the datasets are estimated using Whisper v3-large-turbo (Radford et al., 2022). This collection of datasets was used both as speech-only and interleaved. Similarly to prior work (Nguyen

et al., 2024; Défossez et al., 2024), we additionally include a text dataset during interleaved SLM optimisation. For that we use a subset from RedPajama (Weber et al., 2024), filtered by Gopher filtering rules (Rae et al., 2021). We provide full dataset statistics in Appendix A.1.

This corresponds to a total of $84k$ hours of real speech and $30k$ synthetic ($5.8B$ and $2.2B$ tokens, respectively). We train all SLMs for $\leq 1$ epoch on the speech data, which corresponds to $\leq 2$ epochs of the interleaved data (though the interleaving pattern, i.e., which words are speech, and which are text vary between epochs). Overall, the dataset mixed at equal ratios gets to $\simeq 20B$ tokens equally split between speech only, text only and interleaved data.

**Optimisation.** We follow the efficient SLM training recipe presented by Maimon et al. (2025a), and use their provided toolkit: *SlamKit*[1]. Specifically, we train a TWIST initialised SLM (Hassid et al., 2023), based on different TextLMs, using a cosine scheduler and the datasets described above (including the synthetic sTinyStories dataset). We provide full training hyper-parameters in Appendix A.2.

**Evaluation** in scaling analysis plays an important role, while some studies address loss or perplexity, others use benchmark metrics as well. Furthermore, for interleaved models one could focus on speech only losses and metrics, cross-modal abilities, etc. For the majority of the scaling analysis, we focus on speech-only abilities as evidenced in likelihood of the speech-only validation, as well as evaluation metrics. Specifically, we evaluate both grammatical abilities - with sBLIMP (Yang et al., 2021) and semantic abilities with Spoken StoryCloze (sSC) and Topic-StoryCloze (tSC) (Hassid et al., 2023).

Notice, both sTinyStories, sSC, and tSC were synthesised using a single speaker dataset, specifically using LJ (Ito & Johnson, 2017) voice. This process is aligned with prior work (Hassid et al., 2023; Cuervo & Marxer, 2024). Hence, to assess how well models trained on single-speaker synthetic datasets generalise to other speakers, we construct a multi-speaker version of the spoken StoryCloze benchmark. Specifically, we use Kokoro text-to-speech (TTS) (Hexgrad, 2025) to synthesise the same texts from the original Spoken-StoryCloze and Topic-StoryCloze datasets with four distinct speakers - male and female, British and American accents. Throughout this study we focus on the average performance across speakers, but provide per-speaker analysis in Appendix B.2.

This setup also allows us to separately generate audio prefixes and suffixes, enabling cross-modal evaluation, following the evaluation protocol of Nguyen et al. (2024); Cuervo et al. (2025); Zeng et al. (2024). Essentially, this is much like the uni-modal StoryCloze where we wish to see whether the "real" continuation is assigned a higher likelihood by the SLM, compared to a distractor. However, for the cross-modal setting the prompt is one modality (e.g. text), and the continuation is a different modality (speech). In other words, given a specific text prompt the SLM needs to select the more likely speech continuation. We publicly release these benchmarks.

## 4 Building the Scaffolding for Scaling

To conduct a scaling analysis of SLMs (Section 5), we start by analysing the impact of key design choices which could affect scaling experiments - namely, the use of synthetic data, model family selection, and differences arising from interleaving and TWIST initialisation.

### 4.1 The Effect of Synthetic Data

We begin by analysing the impact of incorporating synthetic data. Following the approach of Cuervo & Marxer (2024); Maimon et al. (2025a), we train interleaved SLMs on both synthetic sTinyStories and real speech data. To assess how synthetic data affects the model's ability to generalise across different speakers, we compare models trained solely on synthetic data, solely on real data, and on a mix of both. All models are initialised from a Qwen2.5–0.5B TextLM (Yang et al., 2024) and trained for 20k steps.

---

[1] https://github.com/slp-rl/slamkit

| Training Data | | Metric | | | | | |
|---|---|---|---|---|---|---|---|
| Real | Syn. | sBLIMP↑ | sSC↑ | MS_sSC↑ | tSC↑ | MS_tSC↑ | Val. CE↓ |
| ✓ | ✗ | 56.77 | 54.94 | 52.66 | 72.15 | 78.93 | **1.96129** |
| ✗ | ✓ | 52.98 | **61.30** | 54.84 | 81.24 | 72.35 | 3.44569 |
| ✓ | ✓ | **56.98** | 59.81 | **54.85** | **81.51** | **81.67** | 1.98267 |

Table 1: Analysing impact of training with/without synthetic data on SLM performance, by training an interleaved Qwen2.5-0.5B based model.

Results provided in Table 1 indicate that models trained solely on synthetic data perform well on metrics involving the same single speaker (LJ)—such as sSC and tSC—but show weaker performance on evaluations involving other speakers, like sBLIMP and multi-speaker tSC. This is also evident from the validation cross-entropy on *real, speech-only* data. While prior research indicates that phonetic speech tokens carry limited speaker information (Sicherman & Adi, 2023), our results suggest that evaluating on a single speaker seen during training may give a misleading impression of model performance. Nevertheless, we see benefit across diverse speakers, by training on a mixture of synthetic and real data, compared to real data only or synthetic data only. This supports the choice of using a mixture of real and synthetic data. In the context of scaling laws, partially replacing real data with synthetic data reduces the real speech data collection needed to train strong SLMs. One could investigate synthetic data with diverse speakers, but we focus on the same single speaker as previous work and specifically Cuervo & Marxer (2024) for consistency.

When considering models trained on real-speech only vs. a mixture of real and synthetic speech data, we note that the validation loss on real speech is lower for the former. We hypothesise that this is due to the validation set, which primarily consists of audiobooks, perhaps not fully representing the model's semantic capabilities. Therefore one should also evaluate using task metrics together with validation loss.

**We therefore suggest** leveraging TTS-generated data from text corpora, providing more semantically coherent input,alongside real speech data to enhance SLM training. In the case of interleaved SLMs, it's important to recognise that validation loss may not always reflect semantic performance accurately. Therefore, evaluation should include a range of metrics, particularly those involving varied and previously unseen speakers.

## 4.2 Model Family Impact

Scaling laws were shown to not always generalise between model families (Choshen et al., 2024a). However, in SLMs initialised from TextLMs, this effect may be more pronounced, as the choice of model family influences not just the architecture but also the weight initialisation—which can play a crucial role. Additionally, the quality of the TextLM used for initialisation can significantly affect interleaving training, since it also involves text tokens.

We conduct an empirical analysis leveraging leading TextLMs ($\leq$ 7B) as a base for interleaved SLMs. Specifically, we use LLama3.2 (Grattafiori et al., 2024), Qwen2.5 (Yang et al., 2024), Gemma2 (Team et al., 2024), SmolLM2 (Allal et al., 2025), Pythia (Biderman et al., 2023), OPT (Zhang et al., 2022), and Bloom (Workshop et al., 2022). Results, provided in Figure 2, suggest that not all model families are born equal. For instance, Pythia160M underperforms OPT125M despite its larger size. In addition, Qwen2.5-0.5B outperforms other, notably larger, models like SmolLM2-1.7B, Bloom-1.7B and OPT1.3B. We note that these results do not always correlate with performance of the base TextLMs. For instance SmolLM2-1.7B outperforms Qwen2.5-1.5B and LLaMa3.2-1B on standard text evaluations (Allal et al., 2025), yet, in our interleaved SLM setup, even the smaller Qwen2.5–0.5B model performs better.

**We therefore suggest** selecting a SLM which shows good performance in smaller scale models (0.5-1B), then using larger versions of the same model. We found Qwen2.5 and LLama3.2 to work well in our tests, and continue most scaling analysis with Qwen2.5 as there are more available model sizes.

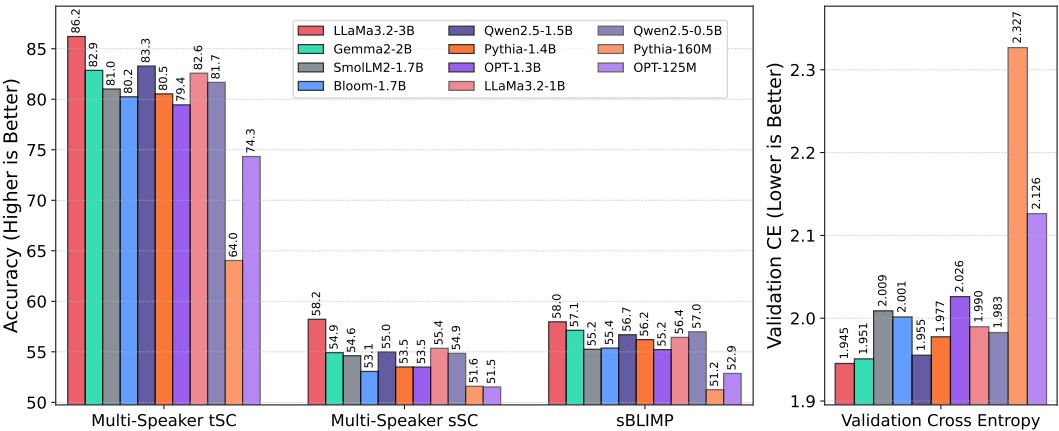

Figure 2: Comparing SLMs trained with the same recipe for 20k steps, from different TextLM initialisations. Models are sorted by parameter count from large to small.

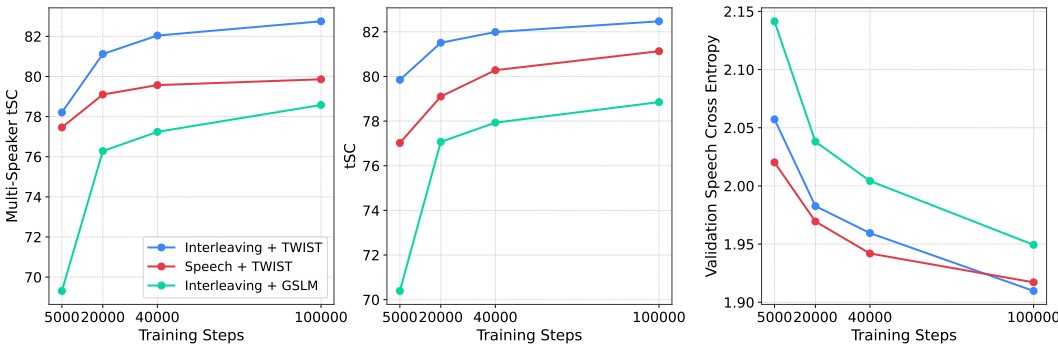

Figure 3: Comparing SLMs based on Qwen2.5-0.5B with interleaving, without interleaving and without TWIST initialisation (denoted GSLM). This helps analyse the impact of these choices on performance and thus on scaling analysis. See Appendix 6 for other metrics.

### 4.3 When Does Text Benefit Speech Modelling?

One might wonder whether or not interleaving could be useful for improving speech only abilities. Nguyen et al. (2024) demonstrated that it does – models trained with interleaving outperformed those trained solely on speech. We follow up this question by asking *"When is interleaving better than speech only?"*

Following the above observation that different model families (and sizes) perform very differently, we focus on a model which showed good performance - Qwen2.5-0.5B. We compare interleaved SLMs trained with the same recipe and initialisation, but varying number of steps. Our results, described in Figure 3, show that for semantic abilities the interleaved model outperformed the speech only model from as little as 5k training steps (∼ 720M tokens). This impact seems to increase with more compute. Other results show that this is not necessarily the case for weaker/smaller TextLMs, e.g. OPT125M, which could require notably more steps for interleaving benefits, see Appendix B.1.

Hassid et al. (2023) showed that initialising SLMs from a pre-trained TextLM could have diminishing returns with regards to training steps/tokens. Intuitively, the longer the model is trained, the less impact the good initialisation has, though not necessarily decaying to zero. We wish to investigate this in the context of interleaved SLMs. We hypothesise that this effect could have an impact on the functional form between $D, N$ and performance. While in existing SLM scaling analysis (Cuervo & Marxer, 2024) the loss in assumed to have

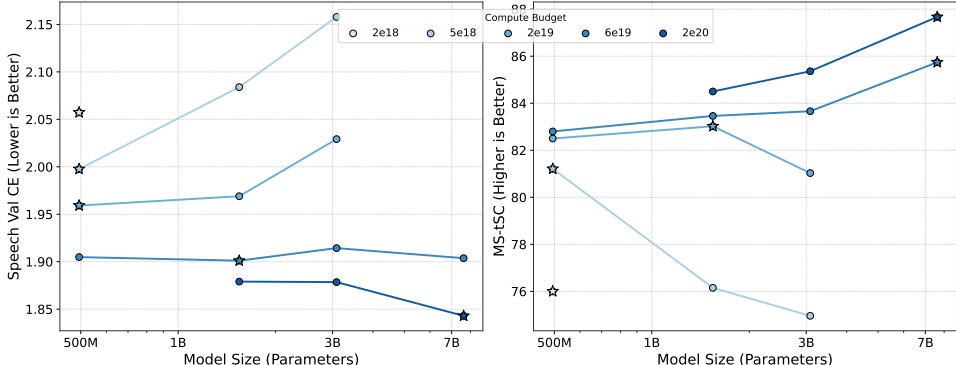

Figure 4: Comparing the loss on speech only of interleaved SLMs of different model sizes trained for specific compute budgets.

an exponential relationship with $\mathcal{D}$, it is not clear this assumption holds in our setup. This training process resembles the addition of a new language to an existing LLM, hence its scaling behaviour might differ significantly. For example, the influence of tokens may not diminish exponentially due to factors like initialisation effects and potential knowledge transfer.

We investigate this by comparing an interleaved SLM initialised from scratch to the common text-initialised version. Results are depicted in Figure 3. We observe that the gap in loss does seem to decrease with more training steps, however the gap in tasks metric is less clear cut (though TWIST initialisation is clearly beneficial). This could mean that the impact of $\mathcal{D}$ on model performance might decay faster, at least for some ranges and initialisations. This could induce a different scaling functional form, but intuitively this would mean that *one should spend more of the compute on parameters*.[2]

**We therefore suggest** always training with interleaving, initialise from a pre-trained TextLM- even when interested in speech-only abilities, given that the base TextLM is good enough and there is a reasonable compute budget (e.g $\simeq 5k$ steps, and $\simeq 720M$ tokens).

## 5 Scaling Analysis

As a functional form is challenging to fit for the complex training dynamics of interleaved SLMs, we follow Hoffmann et al. (2022) in directly fixing the compute budget and training different model sizes. This follows a similar approach to the ISO-flop curve.

While working with pre-trained TextLMs restricts us to fixed model sizes—limiting the granularity of our scaling analysis and making it harder to assess low-compute regimes—we found that analysis based on smaller but lower-quality models (e.g., Pythia) do not generalise well to real-world applications. Therefore, we opt for Qwen2.5 family due to its strong performance (see Section 4.2) and the availability of a reasonably diverse set of model sizes. We name this **S**peech **I**nterleaved **M**odel **S**caling suite - *SIMS* for short.

We fix the compute budget to a certain FLOP count, and then calculate the appropriate number of tokens $\mathcal{D}$ for each model by using $C(FLOPs) = 6ND$ (Kaplan et al., 2020). We use model sizes - 0.5B, 1.5B, 3B, 7B (omitting irrelevant models where applicable), and compute budgets - 2.1$e$18, 5$e$18, 2$e$19, 6$e$19, 2$e$20. We inspect both speech-only validation loss, and a semantic speech metric - multi-speaker tSC.

The results described in Figure 4 show that compute allocated to model parameters should be large. While we lack fidelity to identify the exact ideal model size per-curve, we can identify trends. For instance, when considering semantic metrics for compute $\leq 5e18$ the

---

[2]One could consider the compute used for training the TextLM, however, these models are widely available thus there is not likely a need to train them from scratch.

| Model | Params | C | Multi-Speaker sSC | | | | | Multi-Speaker tSC | | | | |
|---|---|---|---|---|---|---|---|---|---|---|---|---|
| | | | S | S→T | T→S | T | Orig. | S | S→T | T→S | T | Orig. |
| SpiritLM* (L2) | 7B | 7.5e21 | **62.80** | **68.49** | **62.63** | 72.67 | ∅ | **90.01** | **94.74** | **79.77** | **97.33** | ∅ |
| SIMS (L3) | 3.2B | 5.9e19 | 58.23 | 66.10 | 60.56 | **73.01** | 74.40 | 86.22 | 90.37 | 76.31 | 96.26 | 95.88 |
| SIMS (Q) | 1.5B | 2e20 | 55.80 | 60.73 | 58.28 | 66.92 | 71.78 | 83.87 | 76.91 | 71.73 | 92.73 | 95.88 |
| SIMS (Q) | 3B | 2e20 | 57.00 | 61.38 | 58.23 | 65.47 | 73.86 | 85.36 | 75.16 | 71.11 | 88.88 | 97.22 |
| SIMS (Q) | 7.6B | 2e20 | 58.73 | 66.23 | 60.10 | 72.85 | 75.09 | 87.67 | 91.35 | 76.11 | 96.26 | 97.92 |
| SIMS (Q) | 7.6B | 6.86e20 | 60.33 | 67.73 | 60.65 | 72.47 | 75.09 | 88.27 | 93.29 | 75.34 | 95.67 | 97.92 |

Table 2: Analysing the cross-modal semantic abilities of different interleaved SLMs. We also report the performance of the original TextLM before training under Orig. L - indicates LLaMa TextLM initialisation, while Q - is Qwen2.5. *We use the official model weights, which were trained for 75% longer than originally reported by Nguyen et al. (2024).

smallest model is best, but beyond that we see that larger models are better - indicating that they are nearer the optimal. For instance, for $2e20$ FLOPs our best model, by a large margin, is **7B parameters on** 4.2B tokens vs. 463**M parameters on** 72**B tokens** suggested by Cuervo & Marxer (2024). This is also in stark contrast to large, common models like SpiritLM which are trained on a much larger ratio of tokens (e.g 7B model for $\geq 100$B tokens). We also note that only $\sim 50-60\%$ of our tokens are speech, making the trend with regards to speech tokens even more pronounced.

While these results are hard to extrapolate directly to larger compute budgets, they do suggest that a much larger part of the compute budget should be dedicated to model parameters when considering interleaving with good base models. This could also match the intuition from Section 4.3, that the benefit of more tokens decays faster because of TWIST initialisation. We also note that using a larger, better, TextLM not only means more efficient optimisation on the speech data (i.e. the architectural impact), but also a better initialisation (due to a better TextLM).

Next, we analyse the relation between the downstream metric performance of the SLMs in relation to the training compute. We provide the results in Figure 1 and compare them to the estimates of textless-SLMs by Cuervo & Marxer (2024). We also fit an equation for the scaling of each, following the approach by Cuervo & Marxer (2024). However, we note that in our case these might be slightly more noisy, as we are limited with model sizes to existing TextLM sizes, our best model could be further from the actual optimal. Nevertheless, the results clearly show improved performance per-compute as well as a more optimistic trend (the slope for sSC is higher, for tSC they are similar, but results are likely near the saturation limit). These results suggest the feasibility of high semantic abilities in SLMs with existing data resources.

**We therefore suggest** using the largest possible model within a high-quality model family for large scale training (given $\geq 25k$ steps and $\geq 4.5B$ tokens). For smaller scales, e.g $6e19$, we suggest following the results here as a rough guideline.

## 5.1 Comparison with Existing Models

Armed with the above insights we scale up training of our interleaved SLM training and show that it reach comparable performance to leading SLMs while using significantly less compute. Specifically, we train a model based on Qwen2.5-7B for 15B tokens (both speech and text), which is equivalent to a compute budget of $\sim 6.86e20$. We note that, while this is the largest compute budget, it is not the largest amount of tokens we trained on, and it is still under the regime of less than one epoch on the speech-only data.

We begin by evaluating the cross-modal capabilities of several of our models in Table 2, and compare their performance to SpiritLM, which operates with more than an order of magnitude more computational resources. Even though most of our analysis focused on speech only evaluation, we observe a similar trend for cross-modal evaluation tasks. Unsurprisingly, performance is the best for text-only. However, perhaps surprisingly the speech only performance sometimes out-performs cross-modal abilities. We hypothesise that the text modelling abilities are better than speech, yet-cross modality poses a challenge.

| Model | Params. | Tokens | sBLIMP↑ | sSC↑ | tSC↑ | GenPPL↓ | Self-BLEU↓ |
|---|---|---|---|---|---|---|---|
| GSLM (Lakhotia et al., 2021) | 100M | 1B | 54.2 | 53.3 | 66.6 | ∅ | ∅ |
| TWIST-7B (Hassid et al., 2023) | 7B | 36B | 59.0 | 55.3 | 74.1 | 93.74 | 3.06 |
| TWIST-13B (Hassid et al., 2023) | 13B | 36B | 59.2 | 55.4 | 76.4 | ∅ | ∅ |
| SyllableLM (Baade et al., 2024) | 300M | 16B | **63.7** | ∅ | 75.4 | ∅ | ∅ |
| Slam -DPO (Maimon et al., 2025a) | 358M | 16.7B | 58.5 | 58.2 | 80.7 | 67.3 | 3.25 |
| Slam (Maimon et al., 2025a) | 358M | 16.7B | 61.1 | 61.3 | 84.2 | 46.6 | 3.75 |
| AlignSLM (Lin et al., 2024) | 7B | 36 + 158B | 62.3 | 61.1 | 86.8 | ∅ | ∅ |
| Cuervo & Marxer (2024) | 823M | 82B | 61.3 | 56.7 | 78.0 | ∅ | ∅ |
| Zeng et al. (2024) | 9B | ~1T | ∅ | 62.4 | 82.9 | ∅ | ∅ |
| Moshi (Défossez et al., 2024) | 7B | 720B | 58.8 | 60.8 | 83.0 | ∅ | ∅ |
| SpiritLM (Nguyen et al., 2024) | 7B | 100B | 58.3 | 61.0 | 82.9 | ∅ | ∅ |
| SIMS (ours) | 7B | 15B | 59.8 | **66.7** | **88.3** | **37.6** | 4.15 |

Table 3: Comparing SIMS to existing SLMs. Results on the upper block of the table correspond to textless-SLMs, while results on the bottom correspond to joint speech-text SLMs.

Thus for complex enough tasks, e.g. sSC, the better text performance is enough to out-weigh the challenge of cross modality. However, for tSC this is not always the case.

We finally compare our model with leading existing SLMs. We provide the results in Table 3. We observe that our model outperforms existing methods on semantic metrics - sSC, tSC and GenPPL often by a large margin, even models that have gone through preference optimisation phase (e.g., Slam, AlignSLM). Regarding sBLIMP which measures grammatical abilities we see that SIMS outperforms existing interleaved SLMs. However, some textless SLMs trained with DPO, a different tokeniser, or notably more tokens reach better results.

# 6 Related Work

## 6.1 Speech Language Models

Various methods for learning speech and audio discrete representations have been studied. The authors of Hsu et al. (2021) use masked language modelling to construct phonetic representations, while Zeghidour et al. (2021); Défossez et al. (2022); Kumar et al. (2023) trained auto-encoders coupled with vector-quantisation for acoustic representations.

The task of Generative Spoken Language Modelling over such speech representations was first introduced by Lakhotia et al. (2021). This was extended by Borsos et al. (2023), which incorporated both semantic and acoustic tokens to generate more expressive speech. Building on these foundations, many other SLMs have been proposed, each aiming to improve performance or address different aspects (Hassid et al., 2023; Baade et al., 2024; Maimon et al., 2025a; Cuervo & Marxer, 2024). Beyond the stream of SLMs we deal with, speech-aware TextLMs such as (Chu et al., 2023; Tang et al., 2023; Kong et al., 2024) have been proposed, where they train an audio encoder as an adapter to a pre-trained TextLM. SLMs were also used to solve various traditional speech tasks (Wang et al., 2023a;b; Popuri et al., 2022; Inaguma et al., 2022; Kreuk et al., 2021; Maimon & Adi, 2023; Roth et al., 2024). Furthermore, many recent approaches incorporate speech-text in input and output (Défossez et al., 2024; Zhang et al., 2023; Cuervo et al., 2025; Zeng et al., 2024).

## 6.2 Scaling Analysis

Scaling analysis in TextLMs explores how performance improves with more computational resources, mainly model size and training data (Hestness et al., 2017; Clark et al., 2022; Kaplan et al., 2020). Hoffmann et al. (2022) introduced three scaling methods: varying training tokens with fixed model sizes, adjusting model size for fixed training FLOPs and fitting a parametric loss function (scaling law). Muennighoff et al. (2023) extended this work by examining how scaling laws apply to TextLMs in multi-epoch training scenarios. Additionally, Choshen et al. (2024b) analysed over $1,000$ scaling laws, providing practical recommendations for conducting scaling research.

In the field of speech, several studies explored scaling approaches, with a primary emphasis on scaling model size and data for multilingual speech recognition (ASR) (Pratap et al., 2020; Xiao et al., 2021; Pratap et al., 2024). Recently, Chen et al. (2025) investigated scaling laws in ASR, revealing that scaling enhances performance on low-resource languages. Most relevant to our work is Cuervo & Marxer (2024), which investigated scaling laws for textless SLMs.

# 7  Conclusion

In this paper we present the first scaling analysis of Interleaved SLMs and discover that using high-quality TextLMs with partly synthetic data can lead to optimistic scaling capabilities. As part of this we discover that under interleaved training more of the compute budget should be assigned to model parameters (and subsequent initialisation quality) at the expense of training tokens. We also provide practical insights for researchers by highlighting the impact of model families and synthetic data. We encourage the community to further investigate SLM scaling under various settings, and improve training efficiency further.

**Acknowledgements.**   This research work was supported in by ISF grant number 2049/22, and by the NVIDIA Academic Grant Program providing computing resources, A100 and H100 GPUs hours on NVIDIA cloud computing cluster.

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

# A  Detailed Setup

## A.1  Datasets

We use several speech datasets for training. Specifically, we use Libri-Light (Kahn et al., 2020), LibriSpeech (Panayotov et al., 2015), SWC (Baumann et al., 2018), Tedlium (Hernandez et al., 2018), PeopleSpeech (Galvez et al., 2021) and VoxPopuli (Wang et al., 2021). We only take English subsets for all datasets. We use full training splits where provided, otherwise splitting 99% for training. Full dataset sizes are in Table 4.

| Dataset | Number of Hours | Number of Tokens |
|---|---|---|
| Libri-Light (Kahn et al., 2020) | 50$K$ | 3.5$B$ |
| LibriSpeech (Panayotov et al., 2015) | 960 | 67$M$ |
| SWC (Baumann et al., 2018) | 750 | 19$M$ |
| Tedlium (Hernandez et al., 2018) | 1.6$K$ | 110$M$ |
| PeopleSpeech (Galvez et al., 2021) | 7$K$ | 480$M$ |
| VoxPopuli (Wang et al., 2021) | 24$K$ | 1.64$B$ |
| sTinyStories (Maimon et al., 2025a) | 30$K$ | 2.2$B$ |

Table 4: Dataset train set sizes that we use.

## A.2  Hyper-Parameters

We train all interleaved SLMs with the same settings - context length of 2048, AdamW optimiser, with a Cosine decay learning rate scheduler with a warm-up of 1% of training steps and a final minimal learning rate of $5e-5$. We use bfloat16 precision, with flash-attention2 and data packing. We preform gradient normalisation with a norm of 0.5. We balance the modalities - text-only, speech-only and interleaved by mixing samples based on length distribution of each dataset, and not on exact tokens per batch. We use a maximum learning rate of $5e-4$ for most models, but lower this to $3e-4$ for training of compute $\geq 2e20$ as the longer step count meant that the learning rate "shoulder" was high for too long and induced instability in training. For the compute budget $6e19$ we considered both $3e-4$ and $5e-4$ for all 4 models, and used the best for each by validation loss. Specifically this was $3e-4$ for Qwen2.5-7B, and $5e-4$ for all others (albeit not by a big difference).

## A.3  Evaluation

For generative perplexity - we use the first 1000 samples of sSC (correct samples only), and take the first 3 seconds as a prompt. We then generate a maximum of 150 tokens with $top-K=25$, and a temperature of 0.8, which matches Slam (Maimon et al., 2025a). We transcribe the samples using Whisper-large-v3-turbo model (Radford et al., 2023) and measure Perplexity using the Llama-3.2-1B model (Grattafiori et al., 2024).

We created a multi-speaker versions of Spoken-StoryCloze and Topic-StoryCloze. We used the same texts as the original versions introduced by Hassid et al. (2023), based on the text dataset (Mostafazadeh et al., 2016), and generated the speech using Kokoro TTS (Hexgrad, 2025) with 4 different voices: af_bella, bf_emma, am_puck, and bm_george.

# B  Additional Results

## B.1  When Does Text Benefit Speech Modelling?

We provide here additional results in analysing SLM performance. Specifically, in Figure 5 we show the impact of interleaving and TWIST initialisation for OPT125, which is a smaller model compared to Qwen2.5-0.5B used in Figure 3. We note that for this model interleaving under-performs compared to speech-only for all but the largest budget (where they are comparable). We also note that the non-TWIST initialised model (GSLM) closes the gap

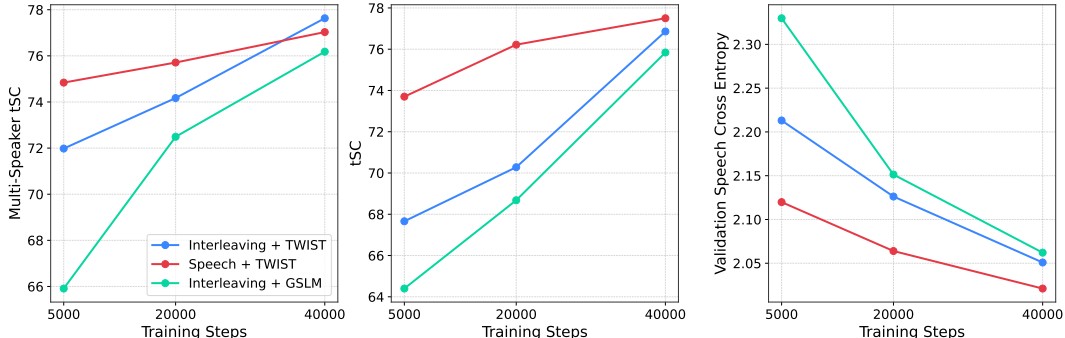

Figure 5: Comparing SLMs based on OPT125M with interleaving, without interleaving and without TWIST initialisation. Comparing to the Figure 3, we can see that OPT125 benefits less from interleaving and TWIST initialisation compared to Qwen2.5-0.5B.

quite substantially, unlike with Qwen. This could further the support the need to use strong TextLM model families to achieve good scaling performance.

## B.2  Diverse Speaker Analysis

In order to analyse how much results vary per-speaker in our suggested multi-speaker benchmark, we report per-speaker metrics for select models. We provide the results in Table 5. Results suggest a slight harm to performance for the British accent, which demonstrates the important of evaluating with diverse speakers. However, the difference in performance across speakers is limited, which shows the robustness of our models.

For evaluation efficiency, we only selected four speakers for the official multi-speaker benchmark. However, in order to further establish to what extent these four speakers represent expected performance on other speakers we select four more speakers. We provide results in Table 6, which show similar trends and scores to the original speakers.

| sSC/tSC | af_bella | am_puck | bf_emma | bm_george |
|---|---|---|---|---|
| SIMS (Q1.5) | 56.23/84.45 | 57.35/83.59 | 54.30/82.68 | 55.32/84.77 |
| SIMS (L3) | 59.27/86.48 | 59.49/86.85 | 57.19/85.03 | 56.97/86.53 |
| SIMS (Q7) | 61.41/88.35 | 62.37/88.62 | 58.52/87.28 | 59.01/88.78 |

Table 5: Model performance (sSC and tSC) for the four diverse speakers used for the multi-speaker versions.

| sSC/tSC | af_heart | am_michael | bf_isabella | bm_fabel |
|---|---|---|---|---|
| SIMS (Q1.5) | 56.81/85.46 | 56.76/85.03 | 55.64/81.24 | 55.53/81.24 |
| SIMS (L3) | 58.90/87.23 | 58.15/87.65 | 56.97/83.86 | 58.20/84.61 |
| SIMS (Q7) | 61.52/90.22 | 61.57/90.06 | 58.52/85.94 | 59.27/85.57 |

Table 6: Model performance (sSC and tSC) for four additional speakers, not used in the multi-speaker version. We select male and female as well as British and American accents. The similarity of results to Table 5 demonstrates that the original four speakers represent a wider population well.

## B.3  Scaling Analysis

We also provide results for additional metrics, corresponding to Figures 3 and 4. Specifically, Figure 6 provides further metrics for analysis of the impact of interleaving and TWIST

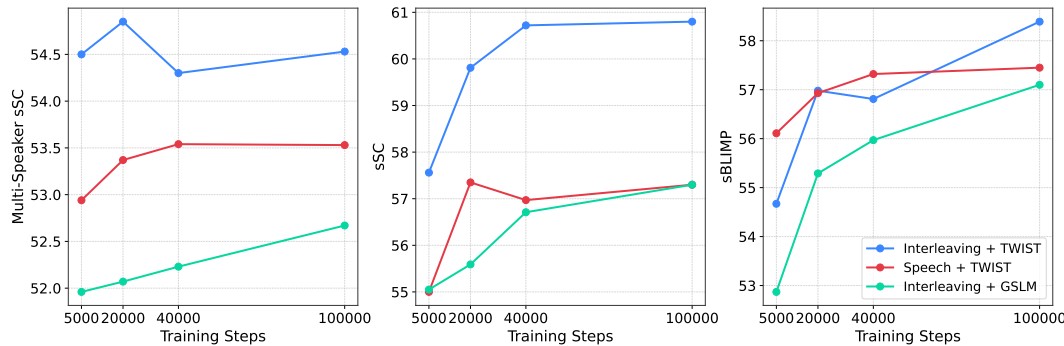

Figure 6: Comparing SLMs based on Qwen2.5-0.5B with interleaving, without interleaving and without TWIST initialisation. This compliments Figure 3, yet results are a bit more noisy, perhaps because they are nearer to random.

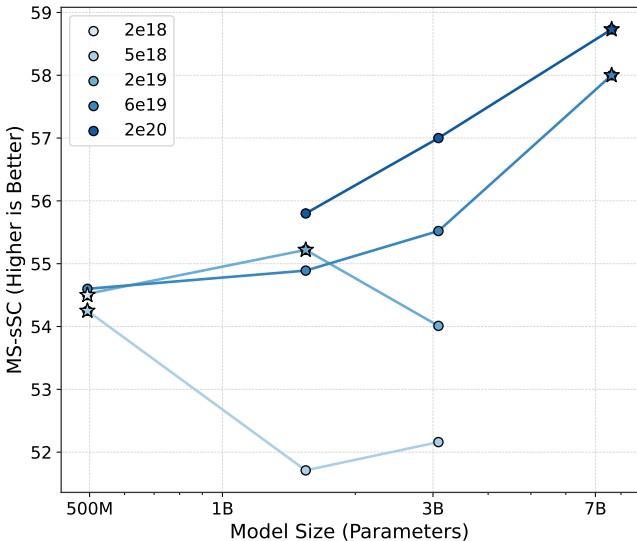

Figure 7: Analysing the scaling properties of interleaved SLMs regarding multi-speaker sSC. This compliments Figure 4, yet results are a bit more noisy, perhaps because they are nearer to random.

initialisation, beyond those provided in Figure 3. Generally, the sSC versions show similar trends to tSC, yet there is much more noise making them harder to interpret clearly. We suspect this could do with the results being fairly low thus more prone to slight noise. Interestingly - for sBLIMP the results correlate more with the loss and show better performance without interleaving. This could be in part to the Interleaved model seeing less speech tokens per-step (as some are interleaved or text), which for low training setup can be detrimental to performance on diverse datasets.

Figure 7 provides scaling analysis like in Figure 4, but for sSC. While showing similar trends to loss and other metrics, there seems to be slightly more variance.

