# OpenReview forum: "Scaling Analysis of Interleaved Speech-Text Language Models"
_colmweb.org/COLM/2025/Conference — COLM 2025_

### Official Review · Reviewer_sDgi · 2025-05-08

**Rating:** 6
**Confidence:** 3
**Ethics Flag:** 1

**Summary:**

This paper presents a scaling analysis of interleaved speech LMs and suggests that more of the compute budget should be assigned to model parameters than training tokens. Along the way, it also shows the effectiveness of using synthetic data. This paper provides focused insights that are useful to the community.

**Questions To Authors:**

- Figure 1. (1) Putting this figure in Introduction before explaining the setup is confusing. (2) From sub-figure sSC and tSC, we cannot conclude that interleaved SLM has "a more optimistic trend."
- L136. Why only the synthetic sTinyStories dataset is used?
- L152. The "cross-modal evaluation" is unclear to me. What does it mean, and how does it work?
- L174. Why "replacing real data with synthetic data reduces the speech data needed to train strong SLMs"? Real data is harder to obtain? You mean "reduce the real speech data needed"?
- Figure 3. More training steps are needed to show longer-term trends. Current data points are not sufficient to conclude that (1) the interleaved model outperformed the speech only model and his impact seems to increase with more compute; (2) comparing interleaved SLM initialised from scratch to the text-initialised version, the gap in loss decreases with more training steps.
- Table 3. What makes SIMS better than other joint speech-text SLMs?

**Reasons To Accept:**

- The scaling analysis of interleaved speech LMs and the proposed training receipt are useful to the community.
- This paper presents comprehensive experiments to justify the analysis setting and conduct the scaling analysis.

**Reasons To Reject:**

- Some concepts are not well explained, and some conclusions are not convincing. See my comments/question bellow.

---

> ### Author Response · Authors · 2025-05-30
> **Author's response**
>
> We would like to thank the reviewer for reviewing our manuscript and providing meaningful feedback. We are happy the reviewer finds our scaling analysis and findings **useful to the community.**. We are also pleased the reviewer finds our paper to **presents comprehensive experiments to justify the analysis setting and conduct the scaling analysis.**
>
>
> **Regarding (1) placing the figure in the introduction, and (2) optimistic trends on the sub-figures:** (1) We wanted to give a quick overview to those familiar with the domain but will reconsider this per your suggestion. (2) We should clarify - results for interleaved SLM are better than speech-only consistently per compute-budget. Furthermore, for sSC the slope of the trend is larger thus “more optimistic trend”. For tSC the results are near the saturation limit (which is probably less than the theoretical 100, and humans and textLMs don’t get perfect scores), thus making the slope harder to address explicitly. We will clarify in the final paper.
>
> **Regarding the usage of synthetic data, sTineStories:** We can see why the phrasing could be confusing, we use a diverse set of real datasets (as specified in Appendix A.1) *and* the synthetic dataset (See Table 1). We wanted to highlight the use of synthetic data as in the context of scaling and data requirements - synthetic data is often easier to collect than real data. We used specifically this synthetic dataset for consistency with prior work [3]. We will clarify for the camera ready.
>
>
> **Regarding cross-model evaluation:** This follows the existing evaluation protocol of [1, 2, 4]. Essentially, this is much like the uni-modal story cloze where we wish to see whether the “real” continuation is assigned a higher likelihood by the SLM, compared to a distractor. However, for the cross-modal setting the prompt is a specific modality (for instance text), and the continuation is a different modality (speech). In other words, given a specific text prompt the SLM needs to select the more likely *speech* continuation. We will clarify for the final manuscript.
>
> **Regarding L174, real and synthetic data in SLM training:** Yes, your correction is in place. We mean that it reduces the *real* data needed which is often harder to obtain (and standardise, and clean), compared with synthetic data which is becoming progressively easier to create at scale with high quality and decent controllability.
>
> **Regarding Figure3:** The trend the interleaved SLMs outperform Speech-only SLMs, for good model families, was prevalent in many preliminary studies as well Figure 3. In Figure 5, we see that for a weaker base model - OPT125, the speech-only model starts with better performance but the interleaved model closes the gap and even overtakes it on some of the metrics. This again shows the interleaved models improve more with larger compute-budget. Figure 5, also shows another example where the TWIST initialisation starts with a larger gap for small steps, but this gap closes with more steps.
>
> **Regarding what makes SIMS better than other joint speech-text SLMs:** Great question! It is somewhat hard to pin-point specific reasons, as many aspects change between models (the approach for joining speech and text, the training data, base model, hyper-parameters). We suspect that many aspects of the training recipe itself - base model, data mixture, optimisation details give some benefit, see Figure 2 and also [5]. However, it is important to clarify that we do not claim that SIMS is *better* than these models, but rather comparable to them. Furthermore, we would like to highlight that we do not focus on creating a state-of-the-art model, but rather to conduct scaling analysis of interleaved SLMs. The fact that most models are a similar size, but notably more tokens, could also suggest that the impact of more tokens decays faster as we hypothesised.
>
> _________________________________________________________________________________________________________
> [1] Nguyen, Tu Anh, et al. "Spirit-lm: Interleaved spoken and written language model." Transactions of the Association for Computational Linguistics 13 (2025): 30-52.
>
> [2] Zeng, Aohan, et al. "Scaling speech-text pre-training with synthetic interleaved data." arXiv preprint arXiv:2411.17607 (2024).
>
> [3] Cuervo, Santiago, and Ricard Marxer. "Scaling properties of speech language models." arXiv preprint arXiv:2404.00685 (2024).
>
> [4] Cuervo, Santiago, et al. "Text-Speech Language Models with Improved Cross-Modal Transfer by Aligning Abstraction Levels." arXiv preprint arXiv:2503.06211 (2025).
>
> [5] Maimon, Gallil, Avishai Elmakies, and Yossi Adi. "Slamming: Training a speech language model on one gpu in a day." arXiv preprint arXiv:2502.15814 (2025).

---

> > ### Comment · Reviewer_sDgi · 2025-06-08
> >
> > Thanks for the detailed responses.
> >
> > - Re Figure 3. The gap between "Interleaving + TWIST" and "Speech + TWIST" is getting larger with more compute measured by Multi-Speaker tSC, but getting smaller measured by tSC until 40K steps. That's why I said that  "more training steps are needed to show longer-term trends."
> >
> > - TWIST is not formally defined, and it's differences with GSLM are not clearly introduced (text-initialized vs. initialized from scratch?).
> >
> > In general, I think the technical contributions are good, but the writing needs to be significantly improved.

---

> > > ### Author Response · Authors · 2025-06-09
> > >
> > > We thank the reviewer again for the detailed response. We believe that we would competently be able to clarify any writing concerns in the final version, based on our fruitful discussion. If the reviewer has any remaining comments or suggestions for further clarifications we are happy to address these as well.
> > >
> > > **Regarding TWIST vs GSLM** - Indeed this refers to initialising weights from a pre-trained TextLM (vs. initialising from scratch).  This is a very common practice in modern SpeechLMs. We will clarify this for the final version of the paper.
> > >
> > > **Regarding more points for longer trends** - We thank the reviewer for the suggestion. Such experiments are quite compute intensive, thus we could not compute them during this discussion period. However, we will make an effort to run the experiments and add them for the final version.

---

### Official Review · Reviewer_T9if · 2025-05-12

**Rating:** 7
**Confidence:** 4
**Ethics Flag:** 1

**Summary:**

This paper examines how language models that combine speech and text (SLMs) grow in capability compared to speech-only models. The key findings show that:

Models trained on both speech and text improve more efficiently as you add computing power than models trained only on speech.
Using a combination of TTS and real speech data helps models better understand different speakers' voices.
Starting with text-trained models before adding speech capabilities (called TWIST initialization) significantly improves understanding of meaning, with greater benefits as more computing resources are used.
These combined speech-text models can match the performance of top speech understanding systems while using fewer resources and less training data than other methods.

**Questions To Authors:**

1. What might cause inaccuracies in your mathematical formulas about how AI models improve with size? Since you only studied a few model sizes and your models might not be ideal, how could these limitations affect your results, and how can we reduce these problems?

2. You found that mixing speech and text during training weakened grammar skills when compared to text-only training. Can you explain how the amount of speech included in training affects performance on different tasks when training resources are limited? What strategies could improve this situation?

**Reasons To Accept:**

The paper thoroughly examines the landscapes of Interleaved SpeechLMs and provides useful observations for building a scalable model.

The empirical results provide several key insights for building a scalable SpeechLLM efficiently.

**Reasons To Reject:**

This work may be relevant to a small sub-community.

---

> ### Author Response · Authors · 2025-05-30
> **Author's response**
>
> We would like to thank the reviewer for reviewing our manuscript and providing valuable feedback. We are happy the reviewer finds our paper **thoroughly examines the landscapes of Interleaved SpeechLMs and provides useful observations for building a scalable model**. We are also pleased the reviewer finds **the empirical results provide several key insights for building a scalable SpeechLLM efficiently.**
>
> **Regarding small subcommunity:** Getting SpeechLMs to listen and speak is gaining popularity across both academic research and industry [9]. Indeed, there are many studies on the best ways to integrate speech and text LMs [1, 5, 9], with many specifically focusing on the interleaving approach [1, 2, 9, 10]. Understanding the scaling dynamics of such approaches helps understand the real world applicability compared to other methods.
>
> **Regarding mathematical formulations of scaling laws for text initialized SLMs:** Indeed conducting scaling analysis when limited to existing pre-trained text model sizes poses a challenge, as we mention in Section 1 (line 47). Most models do not come in very diverse (small) sizes, especially when considering only single, high quality model families. The best, but most unfeasible, way to do this is to train from scratch the base textLMs across diverse model sizes, but this is very costly and out of scope. Another option would be to try to consider different model families which are somewhat similar, for instance use Qwen3 as well as Qwen2.5 which could have slightly different sizes. Our analysis shows a large impact to model family making this possibly impractical. We also note that the mathematical formula not fitting perfectly, is not necessarily a lacking data issue but perhaps a real formulation change. As this scaling analysis is more like adding a new language (and tokens), the scaling properties could be notably different! For instance, the impact of tokens might not decay exponentially because of the impact of the initialisation (further encouraging less steps to prohibit catastrophic forgetting).
>
> **Regarding the ratio of mixing speech and text:** The ideal ratios between speech and text in training are indeed an interesting question. We follow [1, 2] in using equal ratios of speech, text, and interleaving tokens. We focus on the total number of tokens out of a given compute-budget. While interesting, running a proper evaluation of ratios jointly with scaling analysis requires a large compute budget, and we leave that for future work.
> ______________________________________________________________________________________________________________
> [1] Nguyen, Tu Anh, et al. "Spirit-lm: Interleaved spoken and written language model." Transactions of the Association for Computational Linguistics 13 (2025): 30-52.
>
> [2] Zeng, Aohan, et al. "Scaling speech-text pre-training with synthetic interleaved data." arXiv preprint arXiv:2411.17607 (2024).
>
> [3] Lin, Guan-Ting, et al. "Align-SLM: Textless Spoken Language Models with Reinforcement Learning from AI Feedback." arXiv preprint arXiv:2411.01834 (2024).
>
> [4] Cuervo, Santiago, and Ricard Marxer. "Scaling properties of speech language models." arXiv preprint arXiv:2404.00685 (2024).
>
> [5] Défossez, Alexandre, et al. "Moshi: a speech-text foundation model for real-time dialogue." arXiv preprint arXiv:2410.00037 (2024).
>
> [6] Guo, Yiwei, et al. "Recent Advances in Discrete Speech Tokens: A Review." arXiv preprint arXiv:2502.06490 (2025).
>
> [7] Hassid, Michael, et al. "Textually pretrained speech language models." Advances in Neural Information Processing Systems 36 (2023): 63483-63501.
>
> [8] Lakhotia, Kushal, et al. "On generative spoken language modeling from raw audio." Transactions of the Association for Computational Linguistics 9 (2021): 1336-1354.
>
> [9] Arora, Siddhant, et al. "On the landscape of spoken language models: A comprehensive survey." arXiv preprint arXiv:2504.08528 (2025).
>
> [10] Cuervo, Santiago, et al. "Text-Speech Language Models with Improved Cross-Modal Transfer by Aligning Abstraction Levels." arXiv preprint arXiv:2503.06211 (2025).

---

> > ### Comment · Reviewer_T9if · 2025-06-05
> >
> > Thank you for the response. I'm happy to keep my score as is.

---

> > > ### Author Response · Authors · 2025-06-05
> > >
> > > We thank the reviewer again for the time and effort.

---

### Official Review · Reviewer_5iR4 · 2025-05-13

**Rating:** 7
**Confidence:** 3
**Ethics Flag:** 1

**Summary:**

This work examines the question of whether interleaved speech-text LLMs initialized from pre-trained text LLMs show better scaling behavior than textless LLMs. The authors conduct various experiments to identify the association between (a) model family and scaling laws, (b) interleaved versus textless SLMs, and the impact of initializing from a pre-trained text model.

**Questions To Authors:**

- Why does your conclusion from Line 234 follow as a logical consequence of the preceding lines ? That is not very clear.
- How does your model perform on real world speech tasks like speech recognition, translation and language understanding ? How does this compare to other models ?

**Reasons To Accept:**

- This paper is well-written and easy to understand.
- This paper appears technically sound, and has useful advice on scaling LMs. Experimental results appear to corroborate the author's premise.

**Reasons To Reject:**

- The impact of choice of acoustic units on SLMs is not explored, which could be crucial to understanding scaling laws in SLMs.
- It is not clear why a single speaker TTS must be used to build SLMs. As the paper itself mentions, there are multiple multi-speaker TTS models that are available for public use, and hence that part of the analysis is not particularly well motivated. Why would a single speaker TTS trained on LJSpeech be preferred over a multi-speaker TTS model while building SLMs ?
- 4 speakers may not be sufficient on the test set to accurately measure speaker independence for a general purpose SLM.

---

> ### Author Response · Authors · 2025-05-30
> **Author's response**
>
> We would like to thank the reviewer for taking the time to review our paper. We are happy the reviewer found our paper **well-written and easy to understand**, **technically sound, and with useful advice on scaling LMs**.
>
> **Regarding the impact of choice of acoustic units:** This can definitely impact the performance, but most scaling laws papers do in fact on a single tokeniser [4] as the experiments are very costly and there are so many different tokenisers used [7]. We focus on one of the most commonly used tokenizer [1, 3, 8, 9] and more importantly the same one as used for existing SpeechLM scaling laws [4] allowing a direct comparison.
>
> **Regarding using a single speaker TTS for synthetic data:** We should clarify - indeed using diverse TTS could improve performance or robustness even further. We follow existing works in using a single speaker TTS [4] for consistency purposes. The main point was that much of the data can be synthetic leading to less harsh requirements on real data. The exact best characteristics of this synthetic data (e.g diversity) is left for future work.
>
> **Regarding 4 speakers may not be sufficient to test speaker independence:** The four speakers include British Male, British Female, American Male and American Female thus fairly diverse across gender and accent. While this of course does not capture the full diversity of human speech, it does go beyond most common approaches which use a single speaker [1, 2, 3, 4, 8]. Per your suggestion we report here the specific results per-speaker and show that while there are gaps between speakers, the performance does not change drastically. This is likely because HuBERT also removes much of the speaker's information. Additionally, per-your suggestion we randomly sample 4 more speakers and report the results here (left 4 are the original, right 4 are new voices).
>
> | tsc/ssc           | af_bella     | am_puck      | bf_emma      | bm_geoerge   | af_heart     | am_michael   | bf_isabella  | bm_fabel     |
> |-------------------|--------------|--------------|--------------|--------------|--------------|--------------|--------------|--------------|
> | qwen1.5b_it_c2_20 | 84.45/ 56.23 | 83.59/ 57.35 | 82.68/ 54.30 | 84.77/ 55.32 | 85.46/ 56.81 | 85.03/ 56.76 | 81.24/ 55.64 | 81.24/ 55.53 |
> | SIMS_llama3       | 86.48/ 59.27 | 86.85/ 59.49 | 85.03/ 57.19 | 86.53/ 56.97 | 87.23/ 58.90 | 87.65/ 58.15 | 83.86/ 56.97 | 84.61/ 58.20 |
> | Sims_7b           | 88.35/ 61.41 | 88.62/ 62.37 | 87.28/ 58.52 | 88.78/ 59.01 | 90.22/ 61.52 | 90.06/ 61.57 | 85.94/ 58.52 | 85.57/ 59.27 |
>
>
>
>
> **Regarding conclusion from Line 234:** The benefit to performance of increasing $D$ - the number of training tokens (and subsequent number of steps) decays faster when there is TWIST init vs without. This can be seen by the gap closing between models with it and without, as training tokens increase. This was consistent across many preliminary studies and results in Figure 3, and can also make intuitive sense as when D->\inf it is likely that the importance of the initialisation decays. If indeed the impact of D decays faster, so for a given compute budget the relative compute for parameters should be larger. We will clarify this in the final version. This is also evident from Figure 4, that training models with more parameters and less tokens leads to better performance, compared to the speech-only SLM scaling laws.
>
> **Regarding additional evaluations:** We follow the standard evaluation methods used in speech-text interleaving models [1, 2, 3], which include both perplexity based, modelling metrics, and generative metrics. These are also in line with existing SpeechLM scaling laws which we believe is important for comparability of results [4]. We would like to highlight that the SpeechLMs evaluated and analysed in this paper are base models (trained for speech/text continuation and speech-text interleaving), and not Speech-Aware LMs [5] or Instruction tuned SpeechLMs [6]. Hence these models are less suitable for instruct-based evaluation tasks (ASR, TTS, SpokenQA, etc.), as they were not trained for such tasks.

---

> > ### Author Response · Authors · 2025-05-30
> > **Author's response (cont.)**
> >
> > _________________________________________________________________________________________________________________________________________________________________
> > [1] Nguyen, Tu Anh, et al. "Spirit-lm: Interleaved spoken and written language model." Transactions of the Association for Computational Linguistics 13 (2025): 30-52.
> >
> > [2] Zeng, Aohan, et al. "Scaling speech-text pre-training with synthetic interleaved data." arXiv preprint arXiv:2411.17607 (2024).
> >
> > [3] Lin, Guan-Ting, et al. "Align-SLM: Textless Spoken Language Models with Reinforcement Learning from AI Feedback." arXiv preprint arXiv:2411.01834 (2024).
> >
> > [4] Cuervo, Santiago, and Ricard Marxer. "Scaling properties of speech language models." arXiv preprint arXiv:2404.00685 (2024).
> >
> > [5] Tang, Changli, et al. "Salmonn: Towards generic hearing abilities for large language models." arXiv preprint arXiv:2310.13289 (2023).
> >
> > [6] Défossez, Alexandre, et al. "Moshi: a speech-text foundation model for real-time dialogue." arXiv preprint arXiv:2410.00037 (2024).
> >
> > [7] Guo, Yiwei, et al. "Recent Advances in Discrete Speech Tokens: A Review." arXiv preprint arXiv:2502.06490 (2025).
> >
> > [8] Hassid, Michael, et al. "Textually pretrained speech language models." Advances in Neural Information Processing Systems 36 (2023): 63483-63501.
> >
> > [9] Lakhotia, Kushal, et al. "On generative spoken language modeling from raw audio." Transactions of the Association for Computational Linguistics 9 (2021): 1336-1354.

---

> > > ### Comment · Reviewer_5iR4 · 2025-06-07
> > >
> > > I thank the authors for their responses. I accept most of their responses, but still maintain my concerns around the utility of existing evaluations and how they may translate to real world applications.
> > >
> > > I encourage the authors to report variance among speakers in the final paper to support their claim about speaker independence, potentially over a larger set than reported here. However, based on the numbers reported above, I don't have significant concerns.

---

> > > > ### Author Response · Authors · 2025-06-08
> > > >
> > > > We thank the reviewer for the time and useful feedback. Per your suggestion we will include some statistics to further show per-speaker limited variability in results in the final version.
> > > >
> > > > We will re-iterate one final time that chat systems (e.g Moshi [6], GLM4-Voice [2]) which are a leading, real world application examples - also use the same metrics as ours to evaluate their base models and chat models - tSC, sSC, sBLIMP. This could suggest that this correlates well with their downstream usage.

---

### Official Review · Reviewer_LQ6D · 2025-05-13

**Rating:** 6
**Confidence:** 3
**Ethics Flag:** 1

**Summary:**

This paper presents a scaling analysis for Speech-Text Language models. Specifically, the paper pushes beyond the state of the art in the following aspects:

1. The paper observes that under an interleaved setting, SLM scales more efficiently with compute than textless setting, and that more compute should be spent on scaling parameters instead of increasing token counts.
2. The paper examines the impact of synthetic data and TextLM model families in the better scaling of interleaved SLM models. Specifically, it shows that TTS-generated synthetic data and interleaved training with initializations from a pre-trained TextLM are both beneficial for scaling parameters.
3. Taking the insight above, the paper was able to train an interleaved 7B SLM that out-performs previous models with less tokens.

In general, I think this is a potentially impactful empirical work that can facilitate interleaved SLM training in the future, mainly because of the importance of the problem it is trying to tackle. In terms of advancing from the state-of-the-art, the paper picked out some good angles to investigate (synthetic data, model initialization), which I believe a lot of the practitioners in this field are also faced with. The arguments presented in this paper are mostly well-supported with experiments, and presented with good clarity.

My main reservation for this paper is in evaluation. Coming from LLM literature, it is surprising to see that the model is only evaluated in two aspects (sBLIMP and story cloze). I would like to understand better if this is constrained by the availability of the dataset or something else, and how does the authors think this would translate to end-task performance after SFT (e.g. ASR, translation, speech QA etc.)

Another minor comment is that the authors may want to normalize the use of hyphens in L145-153 for "single-speaker" and "multi-speaker". This might have occurred elsewhere in the paper.

**Reasons To Accept:**

1. This is an important problem to investigate in the field -- one should not expect textless SLM scaling law to apply to interleaved training because of the significant difference in data
2. The statements are well-supported with empirical evidence.
3. The presentation quality is high.

**Reasons To Reject:**

I think there's a risk that the evaluation conducted in this paper does not cover enough downstream tasks and may not generalize well to downstream performance of common speech-text LLM applications.

---

> ### Author Response · Authors · 2025-05-30
> **Author's response**
>
> We would like to thank the reviewer for taking the time to review our paper. We are happy the reviewer found our paper **investigating an important** in the field, contianes **well-supported empirical evidence**, and of **high-quality presentation**.
>
> **Regarding additional evaluations:** We follow the standard evaluation methods used in speech-text interleaving models [1, 2, 3], which include both perplexity based, modelling metrics, and generative metrics. These are also in line with existing SpeechLM scaling laws which we believe is important for comparability of results [4]. We would like to highlight that the SpeechLMs evaluated and analysed in this paper are base models (trained for speech/text continuation and speech-text interleaving), and not Speech-Aware LMs [5] or Instruction tuned SpeechLMs [6]. Hence these models are less suitable for instruct-based evaluation tasks (ASR, TTS, SpokenQA, etc.), as they were not trained for such tasks.
> ________________________________________________________________________________________________________
> [1] Nguyen, Tu Anh, et al. "Spirit-lm: Interleaved spoken and written language model." Transactions of the Association for Computational Linguistics 13 (2025): 30-52.
>
> [2] Zeng, Aohan, et al. "Scaling speech-text pre-training with synthetic interleaved data." arXiv preprint arXiv:2411.17607 (2024).
>
> [3] Lin, Guan-Ting, et al. "Align-SLM: Textless Spoken Language Models with Reinforcement Learning from AI Feedback." arXiv preprint arXiv:2411.01834 (2024).
>
> [4] Cuervo, Santiago, and Ricard Marxer. "Scaling properties of speech language models." arXiv preprint arXiv:2404.00685 (2024).
>
> [5] Tang, Changli, et al. "Salmonn: Towards generic hearing abilities for large language models." arXiv preprint arXiv:2310.13289 (2023).
>
> [6] Défossez, Alexandre, et al. "Moshi: a speech-text foundation model for real-time dialogue." arXiv preprint arXiv:2410.00037 (2024).

---

> > ### Comment · Reviewer_LQ6D · 2025-06-03
> >
> > Makes sense. My doubt on how this scaling law maps to real-world end-task performance remains, but I also don't think this should prevent the paper from being accepted. I'll maintain my score.

---

> > > ### Author Response · Authors · 2025-06-04
> > >
> > > We thank the reviewer again for their feedback, and taking the time to respond.
> > >
> > > We will re-iterate one final time that chat systems (e.g Moshi [6], GLM4-Voice [4]) which are a leading, real world downstream usage examples - also use the same metrics as ours to evaluate their base models and chat models - tSC, sSC, sBLIMP. This could suggest that this correlates well with their downstream usage.
> > >
> > > Thanks again!

---

### Decision · Program_Chairs · 2025-07-08

**Decision:**

Accept

**Comment:**

Summary and significance:

This paper investigates the scaling trends for speech language models (SLMs). Previous work has discouragingly shown that SLMs have more unfavorable scaling behavior than text LLMs. This work takes a different approach by interleaving text and speech tokens together, demonstrating that doing so improves the scaling performance of the models. The paper also shows that high quality synthetic data (TTS) can be used to help scale these SLMs.

The main significance of the paper is to highlight a promising new training strategy, interleaved text/speech LMs, as a way to improve the scaling behavior of SLMs.

Pros:

The main discovery of the improved scaling behavior of interleaved text-speech LMs will be impactful for the research community (LQ6D, 5iR4, T9if, sDgi)

The arguments made by the paper are clear and well-supported by experiment (LQ6D, 5iR4)

Cons:

Compared to text-only LLM research, the evaluation tasks used in this paper are limited to only sBLIMP and story cloze (LQ6D)

5iR4 pointed out that the paper's experiments only used one speech tokenizer and a small set of speakers, both of which may change the scaling trends of SLMs.

T9if believes that the work may be relevant more to speech researchers and less to the broader LLM community

sDgi thought several technical details were unclear.

Cons addressed by discussion:

Regarding the limited evaluation criticisms (LQ6D), the authors point out that the sBLIMP and story cloze tasks are the main evaluations used by prior literature on this problem, and that their focus is on the scaling behavior of base models rather than instruction-tuned models. LQ6D acknowledged this rationale makes sense but still was left wanting to see how these base model improvements translated to real world performance.

The authors replied to 5iR4's criticisms about varying the tokenizer and speaker generalization by pointing out that most LLM scaling papers use only one tokenizer because it is too expensive to exhaustively compare many tokenizers. They acknowledge that future work should look at training SLMs with wider speaker variability, but do present additional results measuring the performance of the model on 4 new speakers and show that it is similar to the performance on the training speakers, which mostly alleviates this criticism.

The authors also clarified sDgi's questions and committed to adding these clarifications to the camera ready paper.